# SPARSE ATTENTION ADAPTATION FOR LONG REASONING

**Yizhao Gao**[1,2*†] **Shuming Guo**[3*†] **Shijie Cao**[2‡] **Yuqing Xia**[2] **Yu Cheng**[4] **Lei Wang**[4] **Lingxiao Ma**[2] **Yutao Sun**[5] **Tianzhu Ye**[2] **Li Dong**[2] **Hayden Kwok-Hay So**[1] **Yu Hua**[3] **Ting Cao**[5] **Fan Yang**[2] **Mao Yang**[2]

[1]The University of Hong Kong
[2]Microsoft Research
[3]Huazhong University of Science and Technology
[4]Peking University
[5]Tsinghua University

## ABSTRACT

We introduce SeerAttention-R, a sparse attention framework specifically tailored for the long decoding of reasoning models. Extended from SeerAttention, SeerAttention-R retains the design of learning attention sparsity through a self-distilled gating mechanism, while removing query pooling to accommodate autoregressive decoding. With a **lightweight** plug-in gating, SeerAttention-R is **flexible** and can be easily integrated into existing pretrained model without modifying the original parameters. We demonstrate that SeerAttention-R, trained on just 0.4B tokens, maintains near-lossless reasoning accuracy with 4K token budget in AIME benchmark under large sparse attention block sizes (64/128). Using TileLang, we develop a highly optimized sparse decoding kernel that achieves near-theoretical speedups of up to 9x over FlashAttention-3 on H100 GPU at 90% sparsity.

## 1 INTRODUCTION

Recent reasoning-focused models such as OpenAI o1 Jaech et al. (2024), DeepSeek-R1 Guo et al. (2025), and Qwen3 Yang et al. (2025a) demonstrate that models' capabilities improve significantly through test-time scaling. By generating longer sequences during inference, these models are able to think and reason more effectively before producing an answer. Empirically, longer generations correlate with stronger reasoning performance. For instance, Qwen3-14B Yang et al. (2025a) outperforms DeepSeek-R1-Distill-Qwen-14B Guo et al. (2025) while producing longer responses on average. Similarly, harder benchmarks such as AIME24 require more tokens per generation than easier ones like MATH-500 Hendrycks et al. (2020).

However, deeper reasoning introduces increasing efficiency challenges. Due to the auto-regressive nature of decoding, later tokens must attend to a longer context, increasing compute and memory demands for the KV cache. As a result, the per-token generation cost grows linearly, while the overall generation cost increases quadratically.

Sparse attention offers a promising approach to addressing the long-sequence efficiency challenges. While it has been studied in general language modeling, its application to reasoning models, which require prolonged decoding, remains underexplored. Our experiment using oracle sparsity (Section 4.2) shows that attention in reasoning models is also inherently sparse, activating only a subset of important tokens is sufficient to maintain the model's reasoning capability. The key challenge lies in effectively identifying and leveraging this intrinsic sparsity.

---

*Equal Contribution.
†Work done during internship at Microsoft Research Asia.
‡Corresponding author.

In this work, we extend SeerAttention Gao et al. (2024) to SeerAttention-R, a sparse attention framework aimed to improve the long decoding efficiency of reasoning models. SeerAttention was originally designed to improve prefill efficiency by selectively activating important attention blocks through a lightweight, self-distilled attention gating mechanism at post-training time. SeerAttention-R retains the core design of self-distilled attention sparsity and introduces modifications to support efficient decoding. Specifically, it removes sequence-level pooling of query to accommodate auto-regressive decoding and adopts a shared sparsity design aligned with Grouped Query Attention (GQA) to enhance hardware efficiency. SeerAttention-R can be integrated into any standard transformer-based pretrained model by adding the learnable gate to the attention layer, without fine-tuning original model parameters.

We apply SeerAttention-R to multiple reasoning-focused open-source models, including Qwen3-4B, 8B, 14B Yang et al. (2025a) and DeepSeek-R1-Distill-Qwen-14B Guo et al. (2025), and evaluate them on several reasoning benchmarks: AIME24, AIME25, MATH-500 Hendrycks et al. (2020), and GPQA-Diamond Rein et al. (2024). Since SeerAttention-R only requires training the gating module, the distillation is lightweight with just 0.4B tokens from OpenR1-MATH-220K Face (2025) being sufficient. Across all models and tasks, SeerAttention-R consistently outperforms the Quest Tang et al. (2024) baseline and maintains near-lossless accuracy under a 4k token budget. Notably, the accuracy gap further diminishes as model size increases. More importantly, this learnable approach enables more **coarse-grained sparse attention** (e.g., a block size of 64 or 128), which further reduces the overhead from sparse attention scheme and improve hardware efficiency.

We implement the block sparse flash decoding kernel using both TileLang til and Triton Tillet et al. (2019), and benchmark it on an H100 GPU with FlashAttention-3 (FA3) Shah et al. (2024) as the baseline. Across a range of combination of sequence lengths, batch sizes, and sparsity levels, our TileLang-based kernel consistently outperforms both Triton and FA3. The gains are especially pronounced at large sequence lengths and batch sizes. For example, at batch size 16 and sequence length $\geq$ 32k, our TileLang kernel achieves near-theoretical speedups of up to $8.6\times$ at 90% sparsity over the FA3 baseline, and delivers a $1.7\times$ speedup compared to the Triton counterpart.

## 2 SEERATTENTION-R

### 2.1 A RECAP OF SEERATTENTION

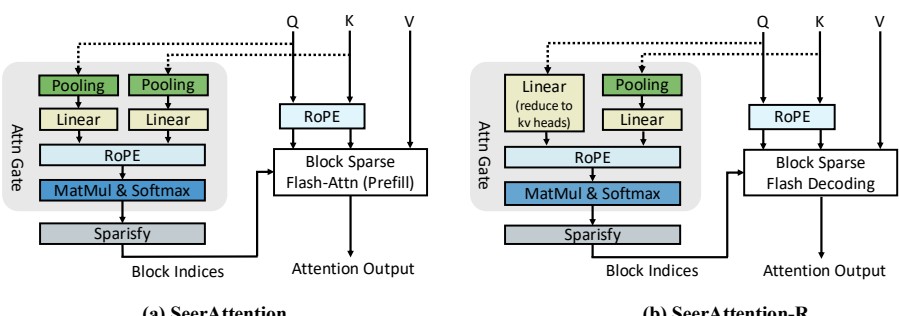

**(a) SeerAttention**          **(b) SeerAttention-R**

Figure 1: SeerAttention (Sparse Prefill) and SeerAttention-R (Sparse Decode). In SeerAttention-R, no sequence dimension compression/pooling operation is applied in Query (Q). Given that modern architectures predominantly use GQA, a linear layer projects the Q from its original number of heads down to the number of KV heads, enabling shared sparsity selection in a GQA group.

SeerAttention Gao et al. (2024) introduces self-distilled *Attention Gate* (AttnGate) that dynamically activates sparse blocks in attention computation for efficient long-context **prefilling**. Figure 1a shows the AttnGate architecture of SeerAttention, where $\mathbf{Q}$, $\mathbf{K}$ tensors are both compressed (pooled) in the sequence dimension per block number of tokens. The compressed $\mathbf{Q}$, $\mathbf{K}$ tensors are then passed through two newly added linear layers, which serve as learnable parameters in the AttnGate. With the following positional embedding, matrix-multiplication and softmax operation similar to standard attention, the AttnGate then generates the 2D block-level attention score estimation. Based on the output, we can selectively activate blocks with higher scores while skipping the rest.

In the distillation process, the AttnGate are trained to mimic the 2D block sparse distribution using the ground truth generated by the original pretrained model. This self-distillation training is efficient as the original model weights are frozen. In this way, it brings accurate sparse attention to pretrained full-attention models without costly fine-tuning or pre-training. Powered by customized block-sparse flash attention kernels, SeerAttention achieves supreme accuracy-efficiency tradeoff in downstream long-context benchmarks.

## 2.2 SEERATTENTION-R: ATTNGATE FOR SPARSE DECODING

This work introduces **SeerAttention-R**, an extension of SeerAttention tailored for the long-decoding phase of reasoning models. The foremost difference of AttnGate design in SeerAttention-R is that it does not apply compression/pooling in the sequence dimension of $\mathbf{Q}$ to accommodate the token-by-token auto-regressive decoding process (shown in Figure 1b).

$$\mathbf{Q_{gate}} = \mathrm{RoPE}\Big(\mathbf{W_{gate}^q}\ \mathrm{reshape}(\mathbf{Q_{nope}}, [...,g \cdot d])\Big), \tag{1a}$$

$$\mathbf{K_{gate}} = \mathrm{RoPE}\Big(\mathbf{W_{gate}^k}\ \mathrm{concat}[\mathrm{P_{max}}(\mathbf{K_{nope}}), \mathrm{P_{min}}(\mathbf{K_{nope}}), \mathrm{P_{avg}}(\mathbf{K_{nope}})]\Big), \tag{1b}$$

$$\mathbf{S} = \mathrm{softmax}(\mathbf{Q_{gate}}\mathbf{K_{gate}}^{\top}/\sqrt{d_{gate}}). \tag{1c}$$

where, $\mathrm{P_{max}}$, $\mathrm{P_{min}}$, and $\mathrm{P_{avg}}$ stand for Max, Min and Average Pooling in sequence dimension, and $g$ is the group size of GQA setting. $d$ and $d_{gate}$ are the hidden dimension of the original model and AttnGate for each head, respectively. $\mathbf{S}$ is the output score of each block from AttnGate. The detailed design are discussed as follows.

**Aggregation of Query Heads for Shared Sparsity in GQA**  Group Query Attention (GQA) Ainslie et al. (2023) is widely used in LLMs to reduce KV cache size. In GQA, the query heads are organized into groups, and each group shares a key-value head. Recent sparse attention works SAAP Mazaré et al. (2025b) and NSA Yuan et al. (2025) show that using identical attention sparsity choices for all queries in a group can improve the efficiency while achieving similar or better performance. In SeerAttention-R, we follow this practice and use a linear layer in the $\mathbf{Q}$ branch of AttnGate to reduce each subgroup of queries to one single head. For example, with 32 query heads and 8 key-value heads (group size $g = 4$), there will be 8 sets of linear weights in shape $[d_{gate}, 4 \times d]$ applying on each group of queries heads, resulting only 8 heads of $\mathbf{Q_{gate}}$. Since we keep the number of heads untouched in $\mathbf{K}$ branch of AttnGate, the final output of AttnGate will be key-value heads, achieving a shared decision of sparsity in a group.

**Pooling-based Compression of Key**  We follow the practice of SeerAttention that uses pooling operations to compress the sequence dimension of $\mathbf{K}$. The kernel and stride size of pooling are both equal to block size, which can also be understood as non-overlapping chunk-level pooling. To mitigate the potential information loss associated with pooling operations, we employ a composition of Max, Min, and Average pooling operations. The outputs from these pooling operations are concatenated prior to being fed into the subsequent linear layer, similar to SeerAttention. The intuition behind this approach is that Max and Min Pooling can effectively capture outlier values, while Average Pooling helps to keep the overall distribution intact.

**Positional Embedding in AttnGate**  In line with SeerAttention, the decode AttnGate utilizes the pre-rope $\mathbf{Q}$ and $\mathbf{K}$ tensors as inputs and reapplies RoPE Su et al. (2024) within AttnGate. Given that the branch is compressed along the sequence dimension, the position index is assigned to the initial token of each block. In our experiment, we found that the use of positional embedding in AttnGate can consistently achieve better accuracy compared to the design without positional embedding.

## 2.3 DISTILLATION/TRAINING

Previous SeerAttention introduces AttnGate distillation method using the ground truth generate by LLM itself in the prefilling phase. The training process is efficient as only the AttnGate are trained. In SeerAttention-R, we extend this method to the decoding scenario by slightly changing the form of the ground truth. Figure 2 shows the overall diagram of the training process.

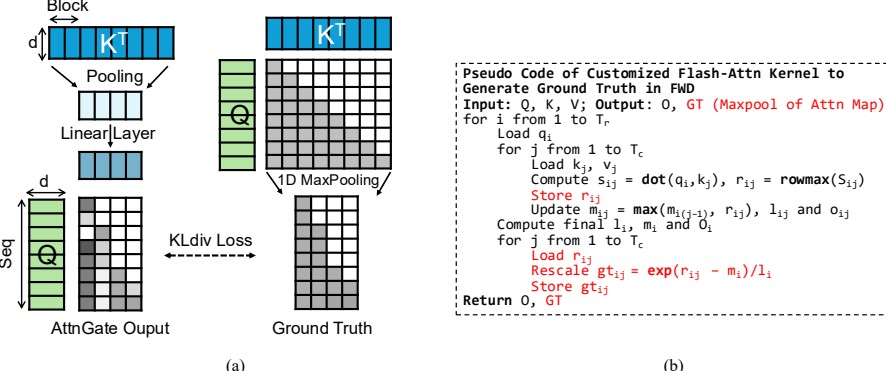

(a)

(b)

Figure 2: Training Diagram and Training Kernel of SeerAttention-R. (a) Self-distillation training of AttnGate in SeerAttention-R. It uses 1D maxpooled attention scores from original model as ground truth to train AttnGate. Query head reduction is not plotted in the diagram for simplicity. (2) Pseudo code of attention forward kernel for training that directly generates ground truth and attention output.

**Ground Truth**    To train AttnGate for the auto-regressive decoding process, we need to adapt the ground truth generation method. Instead of performing 2D maxpooling of attention map in the prefill case, we only do column-wise 1D maxpooling shown in Figure 2a. This corresponds to the decoding AttnGate that does not compress in sequence dimension. Moreover, to accommodate the shared sparsity in GQA, the column-pooled attention map is further maxpooled within each query heads subgroup, resulting in a ground truth with key-value heads. Finally, the ground truth is normalized to sum to 1. We then use the Kullback-Leibler divergence loss Joyce (2011) to train AttnGate in the distillation process.

**Efficiently Obtaining Ground Truth during Training**    Explicitly calculating the full attention map softmax$(\mathbf{QK^T}/\sqrt{\mathbf{d}})$ and then perform the block-level pooling can cost huge GPU memory due to the quadratic complexity. In SeerAttention-R, we also provide an efficient modification of FlashAttention-2 Dao (2023) kernel that directly generates the ground truth along with the attention output. This kernel largely reuses the intermediate results (e.g. block-level rowmax) in Flash-Attention and thus increases the efficiency of the distillation process. The pseudo code is shown in Figure 2b.

## 3    INFERENCE OF SEERATTENTION-R

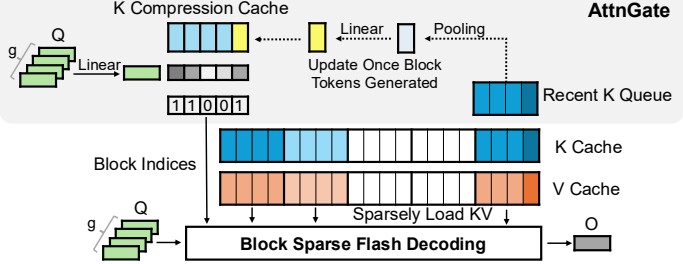

Figure 3: Inference Diagram of SeerAttention-R. During inference, a K Compression Cache is used to cache the compressed key representation in AttnGate to speedup sparse block prediction. This K Compression Cache only updates once per block number of tokens is generated (block=4 in the plots for illustration). As a result, the last block of sequence is always selected to compensate when the compression cache has not been updated yet. $g$ is the group size of GQA.

### 3.1 Sparsify Methods: Token Budget vs Threshold

During training, the AttnGate output $\mathbf{S}$ is distilled to mimic the distribution of the block-wise attention maps from the original model in real-valued (floating-point) form. During inference, important key-value blocks can be selectively activated based on the predictions of AttnGate. In SeerAttention-R, we apply two sparsity methods to convert the soft AttnGate outputs into binary block masks (or block indices). The first method is the *token budget* approach, which is widely adopted in sparse attention methods. Given a fixed token budget, it is first translated into a block budget by dividing the token budget by the block size. The AttnGate outputs are then sorted using a Top-k kernel, where k corresponds to the block budget. While this method introduces an additional Top-k operation, it eliminates the need for a softmax operation in AttnGate. The second method is the *threshold* approach, which simply selects blocks whose scores exceed a given threshold. The threshold method is more self-adaptive as different heads may automatically infer different sparsity ratios. While these two methods involve different trade-offs between efficiency and accuracy, the token budget approach is better suited for direct comparisons with other methods.

### 3.2 K Compression Cache

Similar to KV cache, in SeerAttention-R, we use a *K Compression Cache* to store the compressed representation of K (after pooling plus linear) to speedup AttnGate prediction. Thus, AttnGate does not need to recompute $K$ branch for past seen tokens. The update of K Compression Cache consists of two phases. First, when the sequence length is not a multiple of block size $b$, the new entry of K Compression Cache may not be accurate. During this time, the last block is always activated to eliminate unnecessary accuracy loss. Second, as long as $b$ number of new tokens are generated, the most recent $b$ tokens will pass through the pooling and linear layer and update the K Compression Cache. In this way, the overhead of AttnGate can be minimized.

In practice, SeerAttention-R utilizes a relatively large block size $b$, such as 64, which significantly reduces the overhead of the K Compression Cache. Specifically when $b = 64$, the additional memory required for the K Compression Cache amounts to only 1/128 (<1%) of the original KV cache size. This minimal overhead makes it highly efficient. Moreover, it introduces the possibility of offloading the larger KV cache to CPU or other storage. During inference, only the activated blocks need to be retrieved and transferred back to GPU memory on demand. Alternatively, sparse attention computations can even be performed on heterogeneous resources, such as the CPU, further optimizing memory usage and enabling efficient handling of long-context decoding tasks.

### 3.3 Block Sparse Flash Decoding Kernel

To accelerate decoding under block-sparse attention, we design a specialized kernel that extends the FlashAttention decoding pattern to support dynamic block sparsity in the key/value memory. Our kernel adopts the grid scheduling strategy of flash decoding for GQA, using a three-dimensional launch space over (*batch, heads_kv, num_split*). This design supports concurrent computation across multiple query groups and key/value shards, maximizing block-level parallelism.

Our block sparse version of the decoding kernel takes the activated block indices from AttnGate (shape [*batch, heads_kv, max_selected_blocks*]), which encodes the selected key/value blocks for each group of query heads. During execution, the kernel only traverses the selected indices and thus skips invalid entries, avoiding unnecessary computation and memory access. To improve load/compute balancing across Streaming Multiprocessors (SMs), we partition the key/value blocks along the *num_split* dimension using *max_selected_blocks* rather than the total number of blocks. This strategy ensures a more uniform work distribution in the presence of sparsity-induced irregularity.

On H100 GPUs, our kernel leverages the *wgmma* instructions for better Tensor Core usage by padding the number of query head groups to 64. We implement the kernel using TileLang til, which automatically applies computation optimizations like tiling Zhu et al. (2022), warp specialization and pipelining Cheng et al. (2025), and memory layout optimizations such as tensorization, rasterization and swizzling Wang et al. (2024) based on the target architecture. Additionally, we provide a Triton-based implementation with the same scheduling strategy, allowing for comparative evaluation.

## 4 EXPERIMENTS

### 4.1 EXPERIMENTS SETUP

**Benchmarks, Models, and Baselines** We evaluate SeerAttention-R on three math reasoning benchmarks: the American Invitational Mathematics Examination: AIME24, AIME25, and MATH-500 Hendrycks et al. (2020), as well as GPQA-Diamond Rein et al. (2024). For model evaluation, we select four open-source pre-trained language models with strong reasoning capabilities: Qwen3-4B, 8B, 14B Yang et al. (2025a), and DeepSeek-R1-Distill-Qwen-14B Guo et al. (2025). All models are based on the standard Transformer architecture with Grouped Query Attention (GQA). We compare SeerAttention-R against standard full attention and Quest Tang et al. (2024). Quest is a training-free sparse attention algorithm applied during decoding, employing a query-aware key-value (KV) cache selection strategy. Specifically, Quest estimates the upper bound of attention scores within each KV block (or "page") to select the most relevant blocks. By default, Quest uses a block size of 16, and keeps the first two layers fully dense to minimize error. In Section 4.3, we set the block size to 64 for both Quest and SeerAttention-R, and apply sparse attention to all layers to enable a direct comparison. We also conduct ablation studies to analyze the impact of varying block sizes (Appendix A.1) and incorporating hybrid dense layers (Appendix A.2). Note that SeerAttention-R enables shared sparsity selection within each GQA group, whereas Quest does not. Across all experiments, we set the max output length to 32,768 tokens. While Qwen3 series of models extends this length to 38,912 for AIME24 and AIME25 in their official report, we fix this output length to ensure consistency and fair comparison across all settings. For SeerAttention-R and the full attention baseline, we report average pass@1 accuracy over 64 samples for AIME24 and AIME25, 8 samples for MATH-500, and 16 samples for GPQA-Diamond.

**Training Setup for SeerAttention-R** To distill AttnGate, we use the OpenR1-MATH-220k Face (2025) dataset for training. Importantly, only the AttnGate is trained, and the original model weights remain unchanged. Inputs are packed into sequences of up to 32k tokens with our variable-length Flash-Attention training kernel that also generates ground truth (Section 2.3). Training is performed with a global batch size of 16 for 800 steps on AMD MI300x GPUs, utilizing DeepSeed ZeRO-2 optimization. We use AdamW optimizer and a learning rate of 1e-3 with cosine decay schedule; a higher learning rate is feasible here because only the AttnGate parameters are trained.

### 4.2 ORACLE SPARSE ACCURACY: HOW SPARSE IS ATTENTION IN REASONING MODELS?

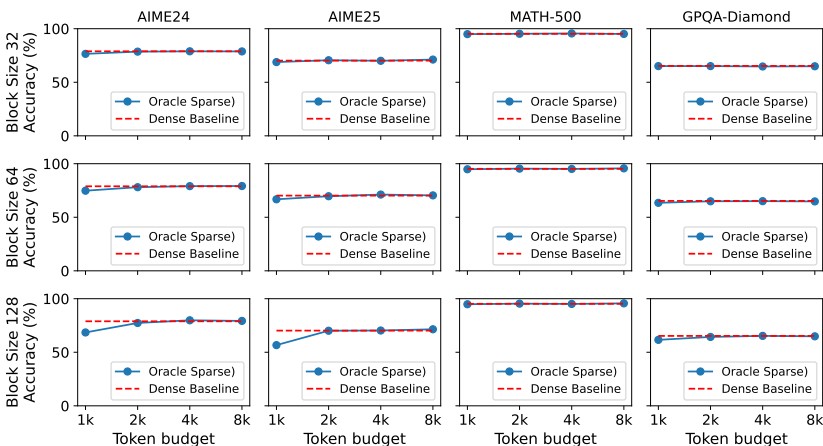

Figure 4: Oracle Sparse Results of Qwen3-14B with block size 32, 64, 128.

In the first experiment, we aim to answer the question: *How sparse is attention in reasoning models?* To investigate this, we employ *oracle block sparse selection*, which uses the same block-level attention score ground truth described in Section 2.3 (column-wise 1D max-pooling of the full attention map) to directly select the top-$k$ KV blocks at each decoding step. This approach serves as an accuracy

upper bound: it reveals how well SeerAttention-R *could* perform if the gating mechanism perfectly predicted which blocks to attend. While this approach requires computing full attention to obtain the ground truth and does not provide any speedup, it allows us to evaluate the accuracy upper bound achievable by SeerAttention-R under ideal sparse selection.

We evaluate Qwen3-14B with three different sparse block sizes: 32, 64, and 128. The token budgets range from 1k to 8k. As shown in Figure 4, using oracle sparsity achieves lossless performance on all tasks when the token budgets reach 2k. For the more challenging AIME24 and AIME25 tasks, some accuracy degradation is observed with 1k token budget, particularly with the largest block size (128). However, this degradation is negligible when using block sizes of 32 or 64. These results indicate that attention sparsity exists in the reasoning process. Based on this, we select a block size of 64 as the default for SeerAttention-R.

## 4.3 RESULTS OF SEERATTENTION-R AND QUEST

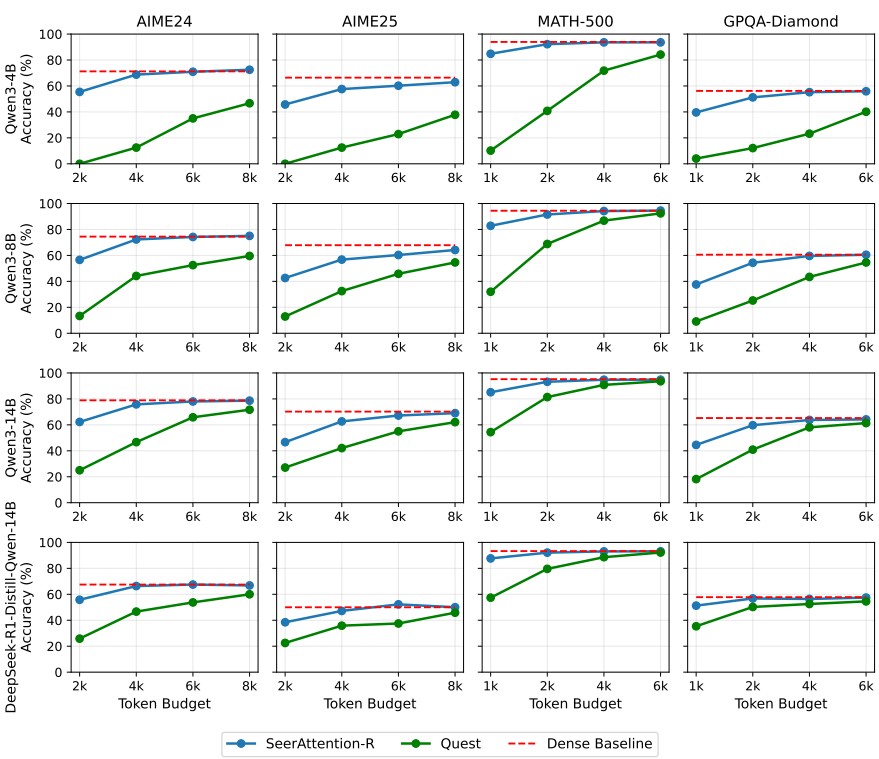

Figure 5: Accuracy Results of Full Attention, SeerAttention-R, and Quest. The Quest sparse configuration is set to be the same as SeerAttention-R for fair comparison, which uses a block size of 64 and sparse attention in all layers.

Figure 5 shows the results of all the models and benchmarks of Full Attention baseline, SeerAttention-R and Quest. As mentioned above, we modify the configuration of Quest to be the same as SeerAttention-R (block size 64 and using sparse attention in all layers). We use token budgets from 2k, 4k, 6k, and 8k for AIME24 and AIME25, and 1k, 2k, 4k, and 6k for MATH-500 and GPQA-Diamond. This is mainly because the typical averaged reasoning length from different benchmark is not the same. For the more challenging AIME24 and AIME25, the averaged generated lengths of these models are around 11k-18k. While for the easier MATH-500 and GPQA, the averaged lengths are reduced to 4k-9k. However, it is critical to note that across all combinations, the maximum generation lengths all reached the 32k token cap, underscoring the consistent demand for efficient long-context processing.

The results show that SeerAttention-R achieves consistently better performance compared to Quest. This trend holds true across every benchmark and computational budget, underscoring the robustness

and effectiveness of SeerAttention-R. For the AIME24 benchmark, SeerAttention-R typically achieves lossless performance with 4k token budget while the previous oracle sparse only requires 2k. This is within expectation as SeerAttention-R is only an approximation of ground truth with much less computation required. However, Quest fails to achieve lossless accuracy even using 8k token budgets under identical setting. For MATH-500 and GPQA-Diamond, the lossless token budgets reduce to 2k for SeerAttention-R while Quest requires around 8k to approach the full attention baseline.

A key trend observed across the results is the relationship between model scale and tolerance for sparse attention. Larger models, such as the 14B variants, exhibit greater robustness to the information loss inherent in sparsity compared to their 4B and 8B counterparts. This phenomenon is particularly pronounced for Quest, where the accuracy gap at lower budgets shrinks significantly as the model size increases. For SeerAttention-R, the effect is also present. The 14B models close the final gap to the dense baseline more easily than smaller models on challenging benchmarks like AIME25. This indicates that as reasoning models continue to scale, the viability of sparse attention methods increases.

In conclusion, the results demonstrate the superiority of SeerAttention-R's self-distilled approach over the training-free heuristics of Quest, especially in the challenging large block size configuration. Previous work LServe Yang et al. (2025b) also mentions the accuracy degradation of Quest over larger block sizes. They resolve this challenge by introducing Hierarchical Paging, a system approach that uses an additional level of block(page) abstraction called virtual logical page, which decouples the sparsity selection page size and physical page size. With SeerAttention-R, we can possibly simplify the sparse attention system design by using a larger block size.

## 4.4 KERNEL SPEEDUP

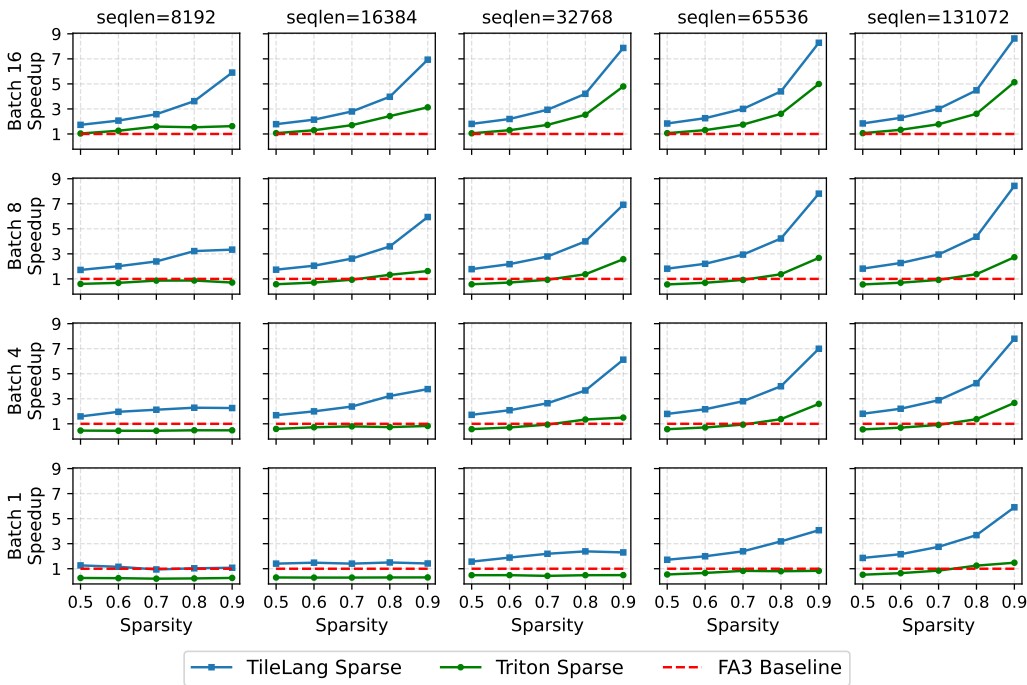

Figure 6: Kernel Speedup of our Block Sparse Flash-Decoding Kernel on H100 GPU. Our TileLang implementation of the kernel achieves higher speedup ratio compared to Triton implementation. For longer sequence length or larger batch size cases, the speedups approach the theoretical upper bound compared to FA3 basline.

This section evaluates our customized block sparse flash decoding kernel described in Section 3.3. We implement the kernel with both TileLang til and Triton and we use FlashAttention-3 (FA3) Shah et al. (2024) as baseline. The experiments are run on Nvidia H100 GPU with different input sequence

lengths (8k to 128k), batch sizes(1 to 16) and sparsity ratios (0.5 to 0.9). For the GQA configuration, we use 64 attention heads with 8 key-value heads, and head dimension 128.

Figure 6 presents the detailed results. Each subplot corresponds to a specific combination of input sequence length (seqlen) and batch size (bs), with the x-axis showing different sparsity ratios and the y-axis indicating speedup. The TileLang implementation consistently outperforms the FA3 baseline and achieves greater speedup than the Triton implementation. In general, the sparse kernel delivers higher speedup when the input sequence length is longer or the batch size is larger. This is expected, as the decoding kernel is primarily I/O-bound. When the KV cache size is sufficient to saturate the bandwidth, such as when bs=16 and seqlen $\geq$ 32k, our sparse kernel achieves near-theoretical speedup (up to $9\times$ at 0.9 sparsity). Even for moderate KV cache sizes, e.g. bs=4 and seqlen=32k, the kernel demonstrates significant speedup (up to $6\times$ at 0.9 sparsity).

## 5 RELATED WORKS

### 5.1 TRAINING-FREE VS. TRAINING-BASED SPARSE ATTENTION

Sparse attention research generally follows two directions: training-free (pre-defined or heuristic) and training-based methods. Training-free approaches adopt static patterns Xiao et al. (2023); Fu et al. (2024); Xiao et al. (2024b) or heuristic-based algorithms Zhang et al. (2023); Tang et al. (2024); Jiang et al. (2024); Yang et al. (2025b); Lai et al. (2025); Chen et al. (2024c); Liu et al. (2024c); Li et al. (2024a); Yang et al. (2024b); Hu et al. (2025); Zhang et al. (2025); Xu et al. (2025); Chen et al. (2025), relying on prior knowledge such as fixed patterns or head characteristics. In contrast, training-based methods integrate sparse attention into models to reduce complexity with minimal accuracy loss. Early work explored local/global/block patterns Child et al. (2019); Beltagy et al. (2020); Zaheer et al. (2020), while recent methods like NSA Yuan et al. (2025), MoBA Lu et al. (2025), ACP Lin et al. (2025b), MiniCPM4 Team (2025) train dynamic sparse modules during pre-training. SeerAttention offers a middle ground, learning sparsity post-training without modifying model weights.

### 5.2 KV CACHE COMPRESSION

KV cache optimization is key for efficient inference, as smaller caches reduce bandwidth and memory costs. Eviction-based methods Ge et al. (2023); Li et al. (2024b); Zeng et al. (2024); Zhang et al. (2023); Liu et al. (2023); Adnan et al. (2024); Chen et al. (2024a); Behnam et al. (2025) permanently remove tokens, risking accuracy loss. Alternatively, dynamic selection methods Tang et al. (2024); Zhang et al. (2024); Hooper et al. (2024); Chen et al. (2025); Liu et al. (2024c); Hu et al. (2025); Cai et al. (2025); Hao et al. (2025); Mazaré et al. (2025a) retain all tokens but select subsets at each step.

### 5.3 OTHER EFFICIENT ATTENTION ALGORITHMS

Beyond sparsity, efficient variants of multi-head attention include GQA Ainslie et al. (2023), MQA Shazeer (2019), latent-based designs Liu et al. (2024a); Zadouri et al. (2025), and cross-layer sharing approaches like YOCO Sun et al. (2024) and CLA Brandon et al. (2024). Linear attention Katharopoulos et al. (2020); Sun et al. (2023); Beck et al. (2024); Yang et al. (2024c); Gu and Dao (2023); Dao and Gu (2024); Peng et al. (2023); Yang et al. (2023) enables parallel training and constant inference memory, though it struggles in long-context reasoning. Hybrid models combining linear and full attention show stronger performance Dong et al. (2024); Li et al. (2025).

## 6 CONCLUSION AND DISCUSSION

This paper introduces SeerAttention-R, a lightweight sparse attention framework that accelerates long decoding in reasoning models. As a plug-in gating module, it integrates into pretrained models without altering original parameters and requires only lightweight training. Despite coarse-grained block sizes, SeerAttention-R preserves near-lossless reasoning accuracy, while its TileLang kernel achieves near-theoretical speedup at high sparsity ratios.

Several challenges remain. Achieving full end-to-end speedup will require integration with inference frameworks (e.g., vLLM Kwon et al. (2023), SGLang Zheng et al. (2024)) and compatibility with

PagedAttention, possibly combined with KV cache offloading Xiao et al. (2024a); Liu et al. (2024c); Chen et al. (2024c); Hao et al. (2025). Another open problem is adaptive sparsity, as the trade-off between accuracy and efficiency varies by task and sequence length. Top-p based sparsity selection Lin et al. (2025a); Chen et al. (2024c) offers one promising direction. Finally, unifying sparse prefill and decoding remains challenging: prefill benefits from parallelism while decoding does not. Approaches such as multi-token prediction Gloeckle et al. (2024); Liu et al. (2024b) or speculative decoding Leviathan et al. (2023) may help align them under a single gating mechanism.

In summary, SeerAttention-R shows that post-training sparse attention can deliver efficiency with minimal accuracy loss, while future work lies in adaptive sparsity, unified prefill/decoding, and system-level integration.

**Limitations.** The current evaluation of SeerAttention-R focuses on mathematical reasoning benchmarks (AIME24, AIME25, MATH-500, GPQA-Diamond) and does not cover other task domains such as long-context retrieval, code generation, or open-domain question answering. While we believe the learned sparsity patterns are general, transferability to non-mathematical tasks remains to be validated in future work. Additionally, our comparison is primarily against Quest; we did not compare with other post-training sparse attention methods for decode that are not evaluated on reasoning tasks such as TidalDecode (Yang et al., 2024a) and MagicPIG (Chen et al., 2024b). We also note a conceptual similarity to the later work DeepSeek Sparse Attention (DSA) (Liu et al., 2025a): both learn attention sparsity via self-distillation. The key difference is the attention variant: DSA targets MLA and achieves token-level selection.

## ACKNOWLEDGEMENTS

This work was supported in part by the National Natural Science Foundation of China (NSFC) under Grant No. U22B2022.

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

# A    APPENDIX

## A.1    BLOCK SIZE FOR SPARSE ATTENTION

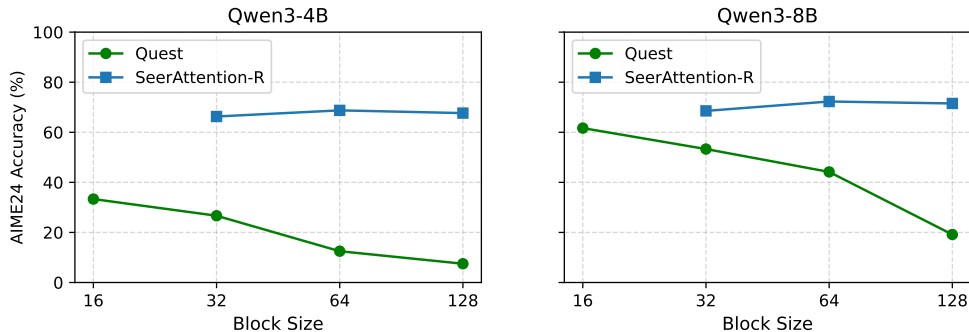

Figure 7: AIME24 results using different block sizes with 4k token budget. SeerAttention-R achieves almost consistent performances on different block sizes. However, Quest gets lower accuracy when block size gets larger. Note that in this experiment, SeerAttention-R enables shared sparsity selection within each GQA group, whereas Quest does not.

The token block size for sparse attention is a critical factor that affects overall system performance. If the block size is too small, it incurs significant overhead in sparse block prediction, including increased computational cost and larger metadata requirements such as compression caches and block indices. While a larger block size can also potentially improve the utilization of GPUs.

Figure 7 presents AIME24 results on the Qwen3-4B and Qwen3-8B models across block sizes ranging from 16 to 128. By default, Quest uses a block size of 16. The results indicate that Quest's performance decreases as the block size increases. However, SeerAttention-R achieves consistent accurate sparse block selection at different block sizes. Remarkably, this robustness lies under the assumption of the additional mask sharing in the GQA group dimension. We excluded a block size of 16 from our experiments due to its inefficiency during both training and inference. It often leads to out-of-memory errors because of the large intermediate attention maps generated during training.

## A.2    HYBRID DENSE ATTENTION IN THE FIRST TWO LAYERS

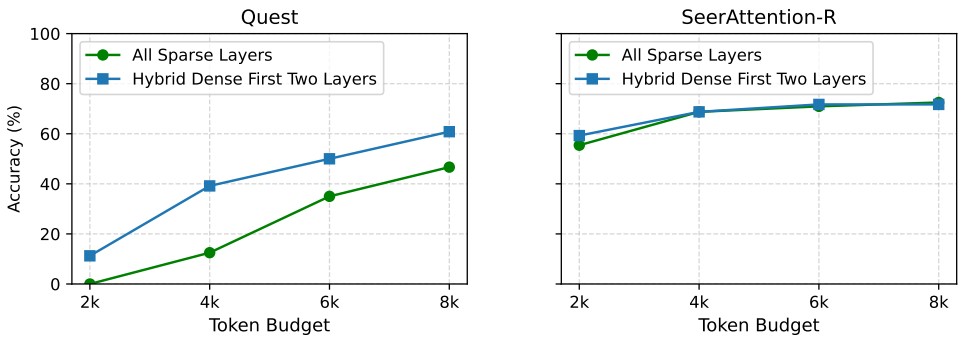

Figure 8: AIME24 results of whether using dense attention in first two layers (Qwen3-4B).

Some post-training sparse attention algorithms employ hybrid dense attention in certain layers to mitigate accuracy loss. By default, Quest applies dense attention in its first two layers. However, for a fair comparison, we evaluate both Quest and SeerAttention-R using purely sparse attention across all layers in previous evaluation. This approach allows us to isolate and analyze the effects of sparse attention without the confounding influence of hybrid attention.

To further investigate the impact of hybrid dense attention, we conduct an ablation study using the Qwen3-4B model on the AIME24 benchmark with a block size of 64. As shown in Figure 8, incorporating hybrid dense attention in Quest yields a significant improvement in accuracy, whereas SeerAttention-R only sees marginal benefits. This difference may be due to the already accurate sparse prediction by SeerAttention-R in the first two layers, reducing the potential gains from hybridization.

## A.3 THRESHOLD VS TOKEN BUDGETS

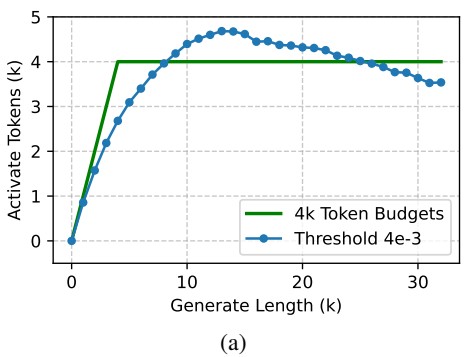
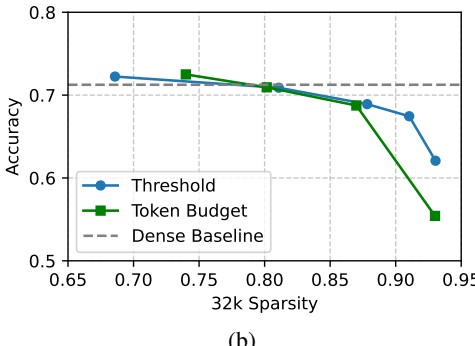

(a)                                              (b)

Figure 9: Threshold vs. Token Budget. Results are obtained using Qwen3-4B models on AIME24 benchmark. (a) Difference of activated tokens distribution of two methods. (b) Sparsity vs Accuracy tradeoff of two methods. Thresholds: 2e-3, 3e-3, 4e-3, 5e-3, 6e-3. Token Budget: 8k, 6k, 4k, 2k.

In SeerAttention-R, we employ two AttnGate sparsification strategies, threshold and token budget, to convert real-valued gate scores into discrete block selections. The token budget method offers an straightforward way to align sparsity and compare with different methods. However, the threshold method is extremely simple to implement and avoids the need of sorting. Figure 9a illustrates the distribution of activated tokens across varying sequence lengths using a threshold of 4e-3 and a token budget of 4K on the AIME24 benchmark with Qwen3-4B model. The token budget approach results in a strict piecewise linear activation pattern, whereas the threshold method yields a smoother, curved distribution. Figure 9b compares the sparsity–accuracy trade-offs of the two methods. The threshold method shows slightly better accuracy in high sparsity region.

## A.4 IMPACT OF SPARSE ATTENTION ON GENERATE LENGTH

Table 1: Qwen3-8B AIME24 Accuracy vs. Reasoning Length.

|  |  | Token Budgets | | | |
|---|---|---|---|---|---|
|  |  | 2k | 4k | 6k | 8k |
| Quest | Accuracy | 13.3 | 44.2 | 52.5 | 59.6 |
|  | Gen. Length(k) | 30.0 | 22.9 | 19.6 | 17.2 |
| SeerAttention-R | Accuracy | 56.6 | 72.3 | 74.2 | 75.1 |
|  | Gen. Length(k) | 19.8 | 16.3 | 15.3 | 15.1 |

We observed that using inaccurate sparse attention (too small budget or low recall) can increase output token lengths in reasoning tasks. Table 1 shows the AIME accuracy and reasoning length using Qwen3-8B model. The baseline accuracy of full attention and the generated length are 74.5 and 15.1 k, respectively. We can see that Quest, and SeerAttention-R with 2k budget cases, all incur much longer reasoning paths compared to full attention. A similar phenomenon has been reported in quantization Liu et al. (2025b), where inaccurate quantization algorithms lead to longer reasoning paths. We believe this effect is universal across different post-training efficiency optimizations of reasoning model, as such methods can introduce errors that accumulate over the long reasoning chains. These additional reasoning steps potentially undermine the original goal of improving efficiency. Therefore, an accurate sparse attention selection algorithm is crucial to mitigate this effect. Another promising approach to eliminate the accumulated errors is to use Rectified Sparse Attention Sun et al. (2025), which periodically performs dense rectification of the KV cache.

A.5   TRAINING BUDGET

Table 2: Training Budgets

| Training Tokens | GPU Hours | | |
|---|---|---|---|
| | Qwen3-4B | Qwen3-8B | Qwen3-14B |
| 0.4B | 10.9 | 12.2 | 18.6 |

As a lightweight distillation process where only the AttnGate parameters are trained, SeerAttention-R is also highly efficient in terms of training. In our experiments, we set the global batch size to 16 and trained for just 800 steps, utilizing DeepSpeed Stage 2 optimization on MI300x GPUs. Each data batch is packed to a sequence length of 32k with our custom variable-length FlashAttention forward kernel, as described in Section 2.3. Table 2 summarizes the GPU hours required for training models of various sizes. Notably, distilling an 8B model requires only 12 GPU hours, demonstrating the efficiency of our approach. Increasing the quantity, quality, and diversity of training data may lead to further improvements.

