# OpenReview forum: "Sparse Attention Adaptation for Long Reasoning"
_ICLR.cc/2026/Conference — ICLR 2026 Poster_

### Official Review · Reviewer_BPwo · 2025-10-30

**Soundness:** 3
**Presentation:** 3
**Contribution:** 2
**Rating:** 4
**Confidence:** 3

**Summary:**

This paper proposes a sparse attention framework designed to accelerate auto-regressive decoding in large language models. The method builds upon SeerAttention, which learns block-level attention sparsity through a lightweight gating network trained via self-distillation. Unlike the original SeerAttention, the proposed approach removes query pooling to make the mechanism compatible with incremental token-by-token decoding. The model further introduces group-wise sparsity sharing aligned with GQA, enabling more efficient decoding with reduced computational overhead. Experiments are conducted to evaluate the proposed method, demonstrating its effectiveness in maintaining model accuracy while improving inference efficiency.

**Strengths:**

- The paper aims to improve the efficiency of auto-regressive decoding in large language models through sparse attention mechanisms. Given the growing importance of long-context inference and low-latency generation, this topic is both relevant and valuable for the ICLR community.

- The main idea is clean and easy to follow. The motivation for each design choice is well explained. The figures and descriptions effectively convey the framework’s structure and its differences from previous work.

**Weaknesses:**

- The proposed approach makes minor modifications to SeerAttention, primarily by removing query pooling to enable auto-regressive decoding. This adaptation is technically reasonable and practically useful, but it does not introduce novel algorithmic ideas or theoretical insights. The contribution feels more like an engineering extension of prior work rather than a fundamentally new method.

- The experimental section could be better organized. Sections 4.3 and 4.4 are clear and easy to follow, but I found Section 4.2 somewhat hard to get. Maybe I missed something, but it is not entirely clear what this part aims to show, and I am also confused about how the “oracle block sparse selection” is defined and implemented. Could the authors provide more explanation or additional details here? My feeling is that this section aims to show that attention in reasoning models is inherently sparse during decoding. Is that correct? If so, this experiment might be better suited as an appendix study or as an initial motivation to support the use of sparse attention in reasoning.

**Questions:**

See the weakness above.

---

> ### Author Response · Authors · 2025-11-19
> **Response to Reviewer BPwo**
>
> ## W1 Novelty of SeerAttention-R
>
> Thanks for raising this concern. We would like to clarify that applying post-training sparse attention to reasoning models is a nontrivial challenge. It is well known that quantization can significantly degrade reasoning quality—often producing shorter outputs or confused reasoning paths—because errors accumulate over long multi-step chains. Sparse attention faces a similar difficulty: if done incorrectly, the model quickly drifts into confusion and low-quality outputs.
>
> Importantly, the recent DeepSeek V3.2 also adopts techniques similar to SeerAttention-R (indexer/AttnGate + attention distillation), demonstrating that this is a practical and effective way to transform a pretrained dense model into a sparse one without incurring prohibitive cost. This further supports our claim that SeerAttention-R provides a useful and novel post-training pathway for sparsifying reasoning models.
>
> ## W2 About the Oracle Sparse Setting
>
> Yes, your understanding is correct. “Oracle sparse’’ means we first compute the full-attention softmax(QK) and then select sparse blocks based on those exact scores. In other words, the selected blocks are guaranteed to be the “correct’’ or “important’’ ones. Under this idealized setting, we can study how many blocks/tokens are actually required.
>
> Our results show that, under the oracle setting, the model only needs around 1k–2k tokens. This provides strong guidance for the experiments, which means that you can compare the token budgets used by a method that achieve lossless performance. For SeerAttention-R it's around 4k-6k tokens. For Quest, it's even can not get close to lossless in 8k.
>
> Interestingly, DeepSeek V3.2 DSA also chooses a 2k token budget, which aligns with our findings. We may move this analysis to the appendix, but we believe the observation is quite insightful: for LLMs, effective attention may only look at 1k–2k tokens per step no matter how long context you give, which is valuable guidance for future pretrained sparse or linear-attention designs.

---

> > ### Comment · Reviewer_BPwo · 2025-11-26
> >
> > Thank you for the clarifications provided in the rebuttal. The additional context on the oracle sparse setting is helpful. I appreciate the authors’ efforts in addressing the questions raised. I have no further comments at this stage.

---

### Official Review · Reviewer_gHf7 · 2025-11-01

**Soundness:** 3
**Presentation:** 3
**Contribution:** 3
**Rating:** 6
**Confidence:** 3

**Summary:**

This paper proposes SeerAttention-R, a sparse attention framework designed for long-sequence decoding in reasoning models. As an extension of SeerAttention, SeerAttention-R retains the core design of learning attention sparsity through a self-distilled gating mechanism while removing query pooling to accommodate auto-regressive decoding. Experiments show that SeerAttention-R, trained on 0.4B tokens, maintains comparable reasoning accuracy on AIME benchmarks, even with large sparse attention block sizes. The optimized sparse decoding kernel implemented using Triton achieves 9× speedup over FlashAttention.

**Strengths:**

1. The paper is well-structured and clearly demonstrates the differences between SeerAttention and SeerAttention-R. Figure 1 contrasts the two methods, allowing to better understand the key insights (removing query pooling, adding shared sparsity design for GQA, etc).

2. The paper provides comprehensive evaluation across multiple models (Qwen3-4B 8B and 14B, DeepSeek-R1-Distill-Qwen-14B) and benchmarks (AIME24/25, MATH-500, GPQA-Diamond).

3. The high training efficiency is interesting. SeerAttention-R requires only 0.4B tokens for lightweight distillation, demonstrating its practical applicability.

4. The paper provides Triton kernel implementation, which is valuable for practical deployment.

**Weaknesses:**

1. The entire paper suffers from citation formatting. Academic papers should properly distinguish between \citep and \citet. The author requires to carefully review every citation in the paper and apply the correct format consistently.

2. The evaluation is restricted to mathematical reasoning tasks (AIME, MATH, GPQA) with training on 0.4B tokens. Generalization to other important domains remains unvalidated, particularly long-context probe tasks and scientific reasoning benchmarks, which are crucial for assessing whether the learned sparsity patterns transfer beyond mathematical reasoning.

3. The paper proposes using a composition of Max, Min, and Average pooling for K compression (Equation 1b) but provides no ablation study comparing this design against alternatives. It would be valuable to see experimental results comparing: (a) using only Max pooling, (b) using only Average pooling, (c) Max+Avg combination, and (d) the full Max+Min+Avg composition. Understanding which pooling components contribute most to performance would strengthen the technical contribution.

4. The experimental comparison is primarily limited to Quest, lacking comparisons with other recent sparse attention methods including NSA, MoBA, and alternative efficient attention mechanisms like linear attention and DeltaNet. Including comparisons with at least a subset of these methods would better position SeerAttention-R's contributions.

Nevertheless, I really like that the authors provide a Triton kernel implementation in addition to TileLang, which significantly enhances the reproducibility and practical utility of this work - this is a major reason for my Weak Accept rating.

**Questions:**

See the Weaknesses described above.

One more question regarding Section "Training Setup for SeerAttention-R": The paper uses the AdamW optimizer with a learning rate of 1e-3 and cosine decay schedule. Could you clarify what weight decay value was used? Additionally, why was 1e-3 chosen as the learning rate, which seems relatively large compared to typical fine-tuning settings?

---

> ### Author Response · Authors · 2025-11-19
> **Response to Reviewer gHf7**
>
> ## W1 Citation Format
> Thanks for pointing out the citation-format issue. We will refine the format accordingly.
>
> ## W2 Evaluation Tasks
> First, GPQA is a scientific reasoning benchmark rather than a math benchmark. Second, we want the paper to be self-contained in its reasoning, since there are many ne challenging problems that have not been carefully studied in prior methods and benchmarks. This is largely because the model’s reasoning path is so long that it easily accumulates errors across steps, which is quite different from earlier long-context probing tasks where you have long-input and short output.
>
> ## W3 Pooling method ablation
> Thanks for the thoughtful question on pooling-method selection. The short answer is that we have already performed this ablation in the prefill-based method for the SeerAttention prefilling attention gate, so we reuse that configuration here. There are also reasons why we believe the composition of max, min, and average pooling is reasonable. First, in Quest shows that the sum of elementwise max between Q*K_minpool, and Q*K_maxpool is an estimation of max attention logits. Also in quantization studeis, people find K vectors have very large outliers in channels. Based on these observations, we believe max min pooling help to capture the extreme value/distribution. Second, works MInference, MoBA, Flexprefill, they use avg pool of Q and K to estimate the block-wise score. Combining max, min, and average therefore allows the gate to use a richer set of signals. In terms of efficiency, as we only periodically update the compress K cache (once block_size steps), the cost remains roughly similar across different pooling choices. In the future, we may also explore pure MLP-based approaches like DeepSeek’s NSA and DSA.
>
> ## W4 Question about comparing other pretrained-based methods.
> Thanks for raising this point. Among the works you listed, SeerAttention-R is the only post-training method, meaning it can convert an already-trained dense model into a sparse one with minimal additional cost. DeepSeek V3.2 DSA adopts a similar idea to convert a dense MLA model to DSA using an indexer and attention distillation. In contrast, NSA, MoBA, DeltaNet, and similar works are pre-trained sparse or linear-attention models, where sparsity is enforced from scratch. Because of this fundamental difference, it is not meaningful to compare them one-to-one with SeerAttention-R at this stage. Below is a summary of different methods that help explain where does SeerAttention-R stand.
>
> | Method Category                   | When Applied                   | Training Cost           | Performance Trend                                    | Example Methods                    |
> | --------------------------------- | ------------------------------ | ----------------------- | ---------------------------------------------------- | ---------------------------------- |
> | **Pre-training Sparse/Linear Attention** | During full model pre-training | High                | Often the strongest given enough training tokens | NSA, MoBA, MiniCPM, DeltaNet etc.           |
> | **Mid-training / Post-training**  | After obtaining a dense model  | Medium → Very Small | **Close to dense baseline**, good cost–performance   | **SeerAttention-R**, DeepSeek V3.2 DSA |
> | **Training-free**                 | None           | None                | Works for specific patterns/heuristics               | Quest, others                      |
>
> ## Question About Learning Setup
> We want to clarify that only the newly added AttnGate modules are trained, and these modules are randomly initialized. Because of this, we choose a larger learning rate. Weight decay has little effect on AttnGate training in our experiments, so we set it to 0 in all final results. If one wishes to further fine-tune the full model together with the trained AttnGate for better performance, then using a smaller learning rate would of course be appropriate.

---

### Official Review · Reviewer_rta5 · 2025-11-01

**Soundness:** 3
**Presentation:** 3
**Contribution:** 4
**Rating:** 8
**Confidence:** 3

**Summary:**

SeerAttention-R is a sparse attention framework for long decoding in reasoning models. It is a modification of SeerAttention: removes query pooling, keeps key pooling and the rest of its gating mechanism. It is lightweight mechanism that integrates into existing models (plug-in gating), needs a fraction of a billion tokens to train (learnable gate). SeerAttention-R maintains reasoning accuracy for large sparse attention block sizes (64/128) and modest reasoning token budgets (e.g. 4K for AIME benchmark). Further, to accelerate decoding under block-sparse attention a special kernel is implemented in TileLang (resulting in close to theoretical speedup). Empirical results (i) on sparse attention demonstrate that SeerAttention-R performs favorably to the approach in Quest (only 4K token to reach the full-attention accuracy) and (ii) on decoding, implementation is faster than one based on Triton.

**Strengths:**

- Important and timely enhancement proposals of SeerAttention for reasoning that are implemented alongside a "matching" decoding kernel using state-of-the-art libraries/languages/tools.

- Empirical results (largely "summarized" in  Figure 6 for (i) and Figure 7 for (ii)) demonstrate clean improvements across all dimensions (token budgets, block sizes, baseline LLMs and benchmark datasets).

- Pragmatic plans for integration with popular inference frameworks.

**Weaknesses:**

- Comparison to only one another sparse attention framework (Quest) could be considered a weak point (on the other hand it keeps the presentation take-aways crisper and easier to consume).

**Questions:**

Quest is a training-free approach, so it does not incur the (small) gate learning (training) overhead, however suffering an accuracy loss for shorter token budgets.
- What about the accuracy/training overhead tradeoff for approaches like those mentioned in Lines 444-446 in Section 5.1?
- How would these compare to SelfAttention-R?

---

> ### Author Response · Authors · 2025-11-16
> **Response to Reviewer rta5**
>
> Thanks for the thoughtful question on comparing SeerAttention-R with other training-based sparse attention methods. One important clarification is that all the cited approaches are **pre-training sparse attention methods**: they train a sparse architecture from scratch, which means they incur the full training cost of large-scale pre-training. In contrast, SeerAttention-R is a post-training method. It converts an already trained dense model into a sparse one with only a very small additional training budget. In general, methods that train on more tokens tend to achieve higher accuracy. This dynamic is similar to quantization: training-aware quantization or native low-bitwidth training from scratch typically achieves stronger results, but at the cost of substantially higher compute.
>
> Back to the question of comparing SeerAttention with these methods. As metioned above, these methods are different pretrained models with different training data and recipes. Therefore, comparing absolute benchmark scores is not meaningful. Still, we found one example: MiniCPM4.1 that provides both dense and sparse variants.
>
> | Model                          | Token Budget         | AIME24 Dense | AIME24 Sparse | AIME25 Dense | AIME25 Sparse |
> | ------------------------------ | -------------------- | ------------ | ------------- | ------------ | ------------- |
> | **MiniCPM 4.1**                | 4k global + 2k local | 83.33        | 80.83         | 73.33        | 72.08         |
> | **SeerAttention-R (Qwen3-8B)** | 6k                   | 74.48        | 74.06         | 67.86        | 64.22         |
>
> This table is not meant as a head-to-head comparison. These models differ in architecture, training corpus, and total training compute. Instead, it illustrates that SeerAttention-R, despite being post-training and low-cost, achieves competitive sparsity–accuracy tradeoffs for challenging reasoning tasks.
>
> We also note that DeepSeek V3.2 DSA recently adopted the same training strategy as SeerAttention-R: post-training, self-distillation–based sparsification for their MLA models, further validating the merit of our method. To help organize the landscape, here is a summary comparing the main families of sparse attention methods:
> | Method Category                   | When Applied                   | Training Cost           | Performance Trend                                    | Example Methods                    |
> | --------------------------------- | ------------------------------ | ----------------------- | ---------------------------------------------------- | ---------------------------------- |
> | **Pre-training Sparse Attention** | During full model pre-training | High                | Often the strongest given enough training tokens | NSA, MoBA, MiniCPM, etc.           |
> | **Mid-training / Post-training**  | After obtaining a dense model  | Medium → Very Small | **Close to dense baseline**, good cost–performance   | **SeerAttention-R**, DeepSeek V3.2 DSA |
> | **Training-free**                 | None           | None                | Works for specific patterns/heuristics               | Quest, others                      |

---

### Official Review · Reviewer_UjW9 · 2025-11-01

**Soundness:** 2
**Presentation:** 3
**Contribution:** 2
**Rating:** 6
**Confidence:** 3

**Summary:**

The paper introduces SeerAttention-R, a sparse attention framework designed to enhance the efficiency of long-context autoregressive decoding in reasoning models. Building upon the original SeerAttention, SeerAttention-R employs a self-distilled gating mechanism to learn attention sparsity, eliminates query pooling to support autoregressive decoding, and offers a lightweight plug-in gating system for seamless integration into existing pretrained models without altering their original parameters. The authors demonstrate that SeerAttention-R, trained on 0.4 billion tokens, maintains near-lossless reasoning accuracy with a 4,000-token budget in the AIME benchmark under large sparse attention block sizes (64/128). Additionally, they develop a highly optimized sparse decoding kernel using TileLang, achieving near-theoretical speedups of up to 9x over FlashAttention-3 on an H100 GPU at 90% sparsity.

**Strengths:**

1. The self-distilled gating mechanism effectively learns attention sparsity, enhancing computational efficiency without significant accuracy loss.
2. The approach of learning attention sparsity through self-distillation is innovative, addressing the challenge of long-context autoregressive decoding in reasoning models.
3. Empirical results show up to 9x speedup over existing methods like FlashAttention-3 on H100 GPUs at 90% sparsity, validating the proposed optimizations.

**Weaknesses:**

1. While the method shows promising results on the three math reasoning benchmark, its performance on other large-scale datasets or tasks remains unaddressed, raising questions about generalizability.
2. The reported speedups are based on H100 GPUs; performance on other hardware configurations is not discussed, which may limit the applicability of the optimization.
3. The paper lacks a direct comparison with other state-of-the-art sparse attention mechanisms, making it difficult to contextualize the proposed method's advantages.
4. There is a lack of detailed ablation studies to isolate the contributions of individual components, such as the self-distilled gating mechanism and the sparse decoding kernel.

**Questions:**

1. How does the self-distilled gating mechanism compare to other attention sparsity learning methods in terms of efficiency and accuracy beyond Quest?
2. What are the limitations of the sparse decoding kernel on hardware other than the TileLang with H100 GPU tensor core, and how does it perform on different infra configurations?
3. How does it performance on other long-context benchmarks with larger token budget?
4. Are there any trade-offs between the level of sparsity and the model's reasoning accuracy, and how can these be balanced?

---

> ### Author Response · Authors · 2025-11-15
> **Response to Reviewer UJW9**
>
> ## W1 Q3: Benchmark
> We evaluted on AIME24, AIME25, MATH-500, and GPQA-Diamond (science reasoning). As mentioned in the title, this paper focus on long-reasoning and these benchmarks are mostly widly-used challenging benchmarks for RL and reasoning models. In fact, for easier questions, these models even do not need to think for very long and attention is not going be the bottleneck anyway. Thus, we believe these benchmarks are appropriate and well aligned with the goals of our study.
>
> ## W2 Q2 About using H100:
> At this time, we only have access to H100 GPUs for experimentation. The TileLang community is actively working on supporting a broader range of hardware platforms, and we expect strong performance on Ampere-class GPUs as well. In fact, Ampere’s minimum tensorcore dimension of M=16 could further improve tensorcore utilization for head-wise parallelism in GQA.
>
> ## W3 Other SOTA Sparse Attention method
> During the development of this work, Quest was, to the best of our knowledge, the state-of-the-art training-free sparse attention method for decoding. LServe also relies on Quest as its underlying algorithm. Several recent works exploring sparsity for reasoning models appeared close to our submission deadline, and we view those as concurrent efforts rather than prior art.
>
> ## W4 Ablation
> We have included a comprehensive ablation study in the original appendix, covering block size, hybrid first-two dense layers, different sparsity-selection algorithms, and the impact of input length. We hope this addresses your concerns. If there are additional ablations you would like to see, could you clarify which specific aspects you are interested in?
>
> ## Q4 Tradeoffs
> All results in our figures inherently reflect a trade-off between token budget and accuracy. For example, in Fig. 5, a smaller token budget corresponds to higher sparsity. You can also refer to Fig. 9(b) that illustrates typical sparsity levels at 32k context length.

---

### Official Review · Reviewer_Maso · 2025-11-02

**Soundness:** 2
**Presentation:** 2
**Contribution:** 2
**Rating:** 2
**Confidence:** 2

**Summary:**

This paper proposes SeerAttention-R extending SeerAttention which was originally designed to improve pre-fill efficiency by sparsely activating important attention blocks with a self distilled gating mechanism at post-training time. The proposed SeerAttention-R keeps the original design choices of SeerAttention but removes sequence-level query pooling to support autoregressive decoding and adopts a shared sparsity design for Grouped Query Attention (GQA). They experimented with different reasoning models and benchmarks and showed good speedups while maintaining the accuracy, by training the gating function on 0.4B tokens.

**Strengths:**

1)  This paper is based on an already proposed sparse attention mechanism. Their contribution is to accommodate the original idea to make it work for autoregressive decoding by removing sequence level query pooling and adapted for recently proposed Grouped Query Attention.

2) They also developed a sparse decoding kernel  using TileLang that achieves up to 9x speedups over FlashAttention-3 on H100 GPU at 90% sparsity.

**Weaknesses:**

1) The results and design is based on GQA shared sparsity. How well does SeerAttention-R transfer to non-GQA like MHA models without the group-sharing trick? Can you provide comparative results on that?

2)  It’s unclear how the reported gains translate in practical inference stacks—e.g., vLLM with FlashAttention—and in settings that use speculative decoding or KV-cache quantitation. Can you quantify the benefits under these configurations?

3)  The proposed method requires additional training, unlike training-free approaches such as TidalDecode [1] and Quest. The paper also didn't a compare with TidalDecode. Can you report results on TidalDecode?

[1] Yang, Lijie, et al. "Tidaldecode: Fast and accurate llm decoding with position persistent sparse attention." arXiv preprint arXiv:2410.05076 (2024).

**Questions:**

Check the weaknesses.

---

> ### Author Response · Authors · 2025-11-15
> **Answer to Reviewer Maso**
>
> ## MHA
> Thanks for your great question about MHA. The “GQA trick” we use is primarily for improving kernel efficiency. Other attention variants, such as MLA, can use even coarser sharing granularity. DeepSeek V3.2 DSA, for example, adopts essentially the same training strategy as ours for MLA model. As for MHA, it is difficult to find an open-source MHA model today because it is generally too inefficient to train and serve. Nevertheless, we expect accuracy to be even better without mask sharing, since removing that constraint increases the expressiveness of the AttnGate.
>
> ## Gain on practical inference stacks
> Thanks for the great question about gains in vLLM-like frameworks. As noted, this paper focuses primarily on the algorithmic side, specifically, improving the accuracy of sparse attention for reasoning models. The same is true for our baselines, Quest and TidalDecode: neither provides native support for vLLM or similar inference stacks, so comparisons are limited to the algorithmic level rather than full system integration.
> Achieving practical end-to-end speedups requires accounting for many system-level factors, including batch size, sequence length, latency constraints, GPU architecture and etc. Fully optimizing the design under these constraints demands substantial engineering effort and is far from trivial. We hope that this work can help motivate the community to explore this space further and contribute to future support for sparse, linear, or hybrid attention architectures in real-world inference systems.
>
> On the other hand, our TileLang kernel can support Page Attention, this is our first step towards incoperting with serving systems like vLLM. However, fully integrating SeerAttention-R also requires revising the AttnGate and K-Compression cache components, which introduces new challenges in how to efficiently manage and update these caches. We are actively working toward addressing these issues.
>
>
> ## TidalDecode
> We have already compared our method with Quest, which is a training-free baseline. TidalDecode is also a strong approach, but its original paper does not evaluate performance on reasoning tasks. According to the follow-up work “Less is More: Training-Free Sparse Attention with Global Locality for Efficient Reasoning,” SeerAttention-R outperforms TidalDecode across all evaluated reasoning benchmarks. However, this comparison is not entirely fair, as both TidalDecode and Less is More require certain layers to run full attention for sparsity selection, whereas SeerAttention-R operates all the layers in sparse attention. In addition, both TidalDecode and Less is More methods require an expensive operation that writes full attention scores out to HBM for sparsity selection, which leads to quadric offchip memory/communication overhead.
>
> In terms of training, we already show in paper that the training cost is very small. Training-based method typically does not require any pre-defined or human observed patterns that can be unreliable. That's also why DeepSeek V3.2 chooses similar training strategy as SeerAttention-R instead of using pre-defined sparse attention strategy.

---

### Meta-Review · Area_Chair_8Ryc · 2025-12-09

**Summary:**

Reviewers were split 3 positive to 2 negative. Their main concerns included limited evaluation/baselines and limited evidence for generalizability to other models and tasks.

**Reviewer Concerns:**

Limited evaluation/baselines
- The authors explained why they believed additional comparisons were not needed. This issue remains outstanding; I don't think reviewers are likely to change their scores in response.

Generalizability to other models/tasks
- Similar to the above concern, the authors explained why they did not think additional comparisons were warranted. This issue also remains outstanding.

**Reviewer Scores:**

Maso: their request for another baselines (TidalDecode) was not addressed, I don't believe they would have changed their score.

UjW9, rta5, gHf7: these reviewers were already positive. Given the short author responses which were mostly clarifications rather than new results, I don't believe they would have changed their scores.

BPwo: the reviewer responded saying that their concerns were addressed by the rebuttal. I interpret this as being willing to raise their score.

While the reviewer concerns remain mostly unaddressed, I lean towards acceptance because 3 reviewers were initially positive, and 1 of the negative reviewers should change their score to positive based on their response to the authors.

---

### Decision · Program_Chairs · 2026-01-26

Accept (Poster)